# Genetic Characterisation and Comparison of Three Human Coronaviruses (HKU1, OC43, 229E) from Patients and Bovine Coronavirus (BCoV) from Cattle with Respiratory Disease in Slovenia

**DOI:** 10.3390/v13040676

**Published:** 2021-04-15

**Authors:** Monika Jevšnik Virant, Danijela Černe, Miroslav Petrovec, Tomislav Paller, Ivan Toplak

**Affiliations:** 1Institute of Microbiology and Immunology, Faculty of Medicine, University of Ljubljana, Zaloška 4, 1000 Ljubljana, Slovenia; monika.jevsnik@mf.uni-lj.si (M.J.V.); mirc.petrovec@mf.uni-lj.si (M.P.); 2Virology Unit, Institute of Microbiology and Parasitology, Veterinary Faculty, University of Ljubljana, Gerbičeva 60, 1115 Ljubljana, Slovenia; danijela.cerne@vf.uni-lj.si; 3National Veterinary Institute, Veterinary Faculty, University of Ljubljana, Gerbičeva 60, 1115 Ljubljana, Slovenia; tomislav.paller@vf.uni-lj.si

**Keywords:** coronaviruses, genetic diversity, HCoV-OC43, BCoV, transmission

## Abstract

Coronaviruses (CoV) are widely distributed pathogens of human and animals and can cause mild or severe respiratory and gastrointestinal disease. Antigenic and genetic similarity of some CoVs within the *Betacoronavirus* genus is evident. Therefore, for the first time in Slovenia, we investigated the genetic diversity of partial 390-nucleotides of RNA-dependent-RNA polymerase gene (RdRp) for 66 human (HCoV) and 24 bovine CoV (BCoV) positive samples, collected between 2010 and 2016 from human patients and cattle with respiratory disease. The characterized CoV strains belong to four different clusters, in three separate human clusters HCoV-HKU1 (*n* = 34), HCoV-OC43 (*n* = 31) and HCoV 229E (*n* = 1) and bovine grouping only as BCoVs (*n* = 24). BCoVs from cattle and HCoV-OC43 were genetically the most closely related and share 96.4–97.1% nucleotide and 96.9–98.5% amino acid identity.

## 1. Introduction

Coronaviruses (CoVs) are widely distributed pathogens associated with respiratory and gastrointestinal diseases in humans and animals [1]. They are the largest enveloped single-strand RNA viruses and belong to the *Coronaviridae* family [2]. Based on the phylogenetic distances of highly conserved domains and according to the new International Committee for Taxonomy of Viruses (ICTV), CoVs are divided into four genera in the *Orthocoronavirinae* subfamily named *Alphacoronaviruses* (divided into 14 subgenera), *Betacoronaviruses* (divided into five subgenera), *Deltacoronaviruses* (divided into three subgenera) and *Gammacoronaviruses* (divided into three subgenera) [3,4,5].

The emergence of Severe Acute Respiratory Syndrome coronavirus (SARS-CoV) (*Betacoronavirus* genus, *Sarbecovirus* subgenus) in 2003, increased interest in hunting for novel CoVs. Before the first SARS epidemic, only two CoVs were described in humans: HCoV-229E (*Alphacoronaviruses* genus, *Duvinacovirus* subgenus) and HCoV-OC43 (*Betacoronaviruses* genus, *Embecovirus* subgenus, *Betacoronavirus* 1 species). Soon after the first SARS CoV epidemic, two additional human CoVs were described: HCoV-NL63 (*Alphacoronavirus* genus, *Setracoronavirus* subgenus) and HCoV-HKU1 (*Betacoronavirus* genus, *Embecovirus* subgenus). These newly discovered HCoVs mostly cause mild upper-respiratory-tract infections, and only in infants, immunocompromised patients; in elderly patients, CoVs can cause more severe respiratory disorders [6,7,8]. In 2012, a new CoV, Middle East Respiratory Syndrome coronavirus (MERS CoV) (*Betacoronavirus* genus, *Merbecovirus* subgenus), was discovered from patients with a mysterious, fatal disease. Both SARS-CoV and MERS-CoV cause severe respiratory diseases [9,10] and are of zoonotic origin [11,12]. As a zoonotic threat, CoV implies the need to monitor CoV associated with domestic animals in contact with humans.

At the end of December 2019, several cases of human patients with viral pneumonia were reported in Wuhan, Hubei Province, People’s Republic of China (PRC) and the novel SARS CoV-2 was discovered, producing fatal coronavirus disease (COVID-19) in around 2% of infected individuals and great public health concern [13]. From January 2020 to April 2021, SARS CoV-2 was spreading through human-to-human transmission throughout the world and producing the largest global pandemic in recent human history with more than one hundred million of infected people. Genetic analysis of the complete genome sequence of SARS CoV-2 revealed a 96.2% recognition rate with bat SARS coronavirus RaTG13 [14,15]. Similar to other coronaviruses, SARS CoV-2 has many potential natural hosts, intermediate hosts and final hosts.

Data regarding CoV infections before the SARS CoV-2 pandemic and strains circulating in Slovenia remain highly limited. From June 2007 to May 2008, 664 specimens were collected from 592 children under six years of age hospitalized at the University Children’s Hospital in Ljubljana and sent for the routine laboratory detection of respiratory viruses. HCoV RNA was detected in 40 (6%, 95% CI: 4.3–8.1%) of 664 samples. Of these specimens, 21/40 (52.5%) were identified as species HKU1, 7/40 (17.5%) as OC43, 6/40 (15%) as 229E, and 6/40 (15%) as NL63 [16].

Bovine coronavirus (BCoV) is inter-species transmissible, and BCoV-like viruses have been detected in several ruminant species and humans. BCoV is distributed worldwide and is associated with neonatal calf diarrhea [17], winter dysentery in adult animals [18], and disorders in the respiratory tract [19]. All BCoV strains characterized from different geographic areas belong to the subgroup initially designated as 2a [20], and according to the new ICTV classification, BCoV belongs to the *Betacoronavirus* genus, genetically closely related to the HCoV-OC43 in *Embecovirus* subgenus and *Betacoronavirus* 1 species. Some previous publications indicated that approximately 90% of the worldwide cattle population has antibodies against BCoV, and a survey on experimental interspecies transmission between wild ruminants, dogs, horses, and calves suggests the importance of BCoV investigation [21,22]. With whole-genome phylogenetic analysis, 83 BCoV strains were classified into two major genotypes (European and American types); moreover, the European and American types were divided into eleven and three genotypes, respectively [23]. BCoV can significantly economically impact the veterinary industry [24], as other coronaviruses can in other animals [25,26].

Nasal swab samples were collected in Slovenia between 2012 and 2014, from twenty-eight herds from 133 affected live cattle that were clinically suffering from symptoms of respiratory disease, and 12.3% of the tested animals were detected as BCoV positive using the real-time PCR method, confirming the regular circulation of this virus in Slovenian cattle herds [27].

Within the *Betacoronavirus* genus, BCoV shares a global nucleotide identity of 96% with human coronavirus HCoV-OC43. Vijgen and co-workers demonstrated, using molecular clock analysis, that HCoV-OC43 has a zoonotic origin and was transmitted from bovine to human around 1890 [28]. CoVs are unique among RNA viruses because of their replication and transcription mechanism; therefore, CoVs are characterized by a high potential of evolution, adaptation, and interspecies jumping [29]. HCoV-OC43 is the most common human coronavirus and has high genetic diversity. Five genotypes of HCoV-OC43 (A to E) have been identified, and Genotype D was dominant between 2004 and 2012 [30,31].

According to previously observed antigenic and genetic similarity, this study provides the first genetic comparison of 66 HCoVs and 24 BCoV circulating strains collected between 2010 and 2016 in Slovenia.

## 2. Materials and Methods

### 2.1. Clinical Specimens

From 2010 to 2016, 16,732 human nasopharyngeal swab samples were collected from patients (female:male = 1:1.14) with acute respiratory tract infections and admitted to University Medical Centre Ljubljana. All nasopharyngeal swabs were sent to the laboratory of the Institute of Microbiology and Immunology for the routine detection of respiratory viruses. The study protocol was approved on 15 March 2016 for human samples by the National Medical Ethics Committee of the Republic of Slovenia (No. 0120-110/2016-2).

From 2010 to 2016, a total of 133 nasopharyngeal swabs from affected live cattle with respiratory illness and 84 from dead cattle with pneumonia were collected and included in the study. Live and dead cattle were from category between three-month- and one-year-old animals. Ethics approval for testing animal specimens was not needed since samples were primarily taken for routine diagnostic surveillance by the local veterinary specialists.

### 2.2. Sample Preparation, Nucleic-Acid Extraction, and Real-Time RT-PCR

Human nasopharyngeal swabs were collected using flocked-tip swabs and transported to the laboratory in a Copan universal transport medium (UTM-RT) system (Copan Italia, Brescia, Italy). Total nucleic acids were isolated from 190 µL of each human nasopharyngeal swab using a total nucleic acid isolation kit on a MagNa Pure Compact instrument (Roche Applied Science, Mannheim, Germany), according to the manufacturer’s instructions. An additional 5 µL of Equine herpesvirus 1 and Equine arteritis virus isolates were added to all samples for external DNA and RNA control and were detected in separate duplex PCR reactions with other targets [32,33].

All four HCoVs (229E, OC43, NL63, and HKU1) and all other respiratory viruses, including respiratory syncytial virus (RSV), human rhinoviruses (HRV), human metapneumovirus (HMPV), human bocavirus (HBoV), adenoviruses (AdVs), parainfluenza viruses 1-3 (PIV 1-3), enterovirus (EV) and influenza viruses A and B (Flu A-B) were detected by using one-step real-time RT-PCR assay in a Step-One Real-Time PCR system (Applied Biosystems, Carlsbad, CA) [34,35,36,37,38,39,40,41]. A total 5 µL of total nucleic acid was added to 15 µL of reaction mixture including 2 X Reaction Mix, SuperScript^®^ III RT/Platinum^®^TaqMix (Invitrogen, Carlsbad, CA, USA). The cycling conditions were universal for all tested respiratory viruses: 20 min at 50 °C, 2 min at 95 °C, and 45 cycles of 15 s at 95 °C and 45 s at 60 °C.

From 133 affected live cattle (collected from 24 different cattle herds) with symptoms of respiratory disease, nasopharyngeal swab samples were collected into sterile swabs (Sigma Virocult^®^, MW 951S, Leicester, UK). From 84 dead cattle with pneumonia and/or diarrhea originated from 76 different cattle herds, 10 cm^3^ lung tissue and/or feces samples were collected. About 1 cm^3^ of the samples was homogenized in dilution 1:10 in RPMI-1640 (Gibco, Life Technologies Inc., Grand Island, NY, USA) and stored at <−15 °C until testing. Total RNA was extracted from 140 µL of homogenate using a commercial kit for RNA extraction (QIAamp^®^ Viral RNA Mini Kit, Qiagen, Hilden, Germany) according to the manufacturer’s instructions. Individual samples were tested using a commercial real-time PCR method, detecting specific nucleic acids of seven different respiratory pathogens, including the detection of endogenous internal positive control (IPC) for controlling the efficiency of extraction and the absence of inhibitors in individual samples. Samples were tested on a 96-tube microplate. On each microplate, the positive controls for all tested pathogens were included. A commercial TaqMan^®^ real-time PCR kit for the detection of seven major ruminant pathogens (LSI VetMAX™ Screening Pack–Ruminants Respiratory Pathogens, LSI, Lissieu, France), which allows the simultaneous detection of the *Micoplasma bovis*, *Histophilus somni*, *Pasteurella multocida*, *Mannheimia haemolytica*, BCoV, bovine respiratory syncytial virus (BRSV), and Bovine parainfluenza 3 (PI-3) was used as previously described [27]. The amplification was performed using an Mx3005P real-time PCR machine (Stratagene, San Diego, CA, USA). The fluorescent signal was detected after each annealing, and the results were presented as a cycle threshold value for individual samples. Analysis of real-time amplification curves was performed using commercial thermal cycler system software; to determine fluorescence baselines, an “auto baseline” was used.

### 2.3. RT-PCR for Coronaviruses and DNA Sequencing

From previously recognized CoV positive samples, a 440-bp-long fragment of the RNA dependent RNA polymerase (*RdRp*) gene was amplified using the primers and protocol as described by Stephensen [42]. The RT-PCR was performed by using a One Step RT-PCR Kit (Qiagen, Holden, Germany). The amplified products were detected via agarose gel electrophoresis and sequenced by Sanger sequencing, as described in a previous publication [43].

### 2.4. Data Analysis

Nucleotide sequences, obtained from the RT-PCR products, were assembled using the DNASTAR software (version 5.05) and compared to the known sequences of the RdRp gene of coronaviruses from the GenBank database, using the Basic Local Alignment Search Tool (BLAST) program; 390-bp-long sequences were aligned using the ClustalW algorithm and with the reference sequences of CoVs from Genbank, including the most closely related sequences of BCoVs and HCoVs. A maximum-likelihood phylogenetic tree was constructed on the GTR + G + I model using MEGA 6.06 software [43]. Genetic relationships among all the included CoVs were calculated with branch statistics using the bootstrap analysis of 1000 replicates.

## 3. Results

From a total of 16,732 tested human samples, 976 (5.8%) were detected as HCoV-positive, and within CoV-positive samples, 523 (53.6%) were negative for other human respiratory pathogens (RSV, HRV, HMPV, HCoVs, HBoV, AdV, PIV, EV, Flu A, and FluB), as tested with real-time RT-PCR methods. The median age of HCoVs-positive patients was 11 years (14 days to 99 years).

From 133 tested live cattle samples, 16 (12.0%) were CoV positive, while from 84 dead cattle, 10 (11.9%) samples were BCoV-positive by real-time RT-PCR, respectively. None of seven respiratory cattle pathogens (*M. bovis*, *H. somni*, *P. multocida*, *M. haemolytica*, BCoV, BRSV and BPI-3) was detected in 30 (13.8%) of the tested bovine clinical samples. The BCoV positive samples were detected in cattle from the age of three months up to one year.

To determine the overall genetic diversity, 66 HCoVs and 24 BCoV have been included in sequencing and phylogenetic analysis. A phylogenetic tree of a 390-bp fragment of the polymerase gene with only representative sequences was prepared, including available sequences from the GenBank database (Figure 1). The phylogenetic analysis shows that the determined Slovenian CoV strains from this study are classified into four different previously determined species, bovine grouping only as BCoVs (*n* = 24) and into human HCoV-HKU1 (*n* = 34), HCoV-OC43 (*n* = 31) and HCoV 229E (*n* = 1).

Thirty-four Slovenian HCoV-HKU1 belong to HKU1 with 97.2–100% nucleotide identity to each other and were further divided into two lineages with 99.7–100% identity within each group and from 97.2–97.7% nucleotide identity between these two lineages. The first lineage, representing Slovenian HKU1/SLO-39995/2013, KX059693, has 100% nucleotide identity with strain HCoV/KENYA/001/2010 (KP112150) and 28 other HKU1 sequences in GenBank from China and Australia. The second lineage, representing Slovenian HKU1/SLO/20580/2010, KX059667, has 100% nucleotide identity to strain CU-H2238/2010 (JX513213) from Thailand and fifty other HKU1 sequences from China, USA and Brazil (Figure 1).

Thirty-one Slovenian HCoV-OC43 belong to OC43 with 98.7–100% nucleotide identity and were further divided into three lineages with 98.7–99.2% nucleotide identity between them. The first lineage (representing Slovenian OC43/SLO/14041/2010, KX059662) has 100% nucleotide identity to HCoV-OC43/UK/London/2011 strain (KU131570) and 18 other sequences in GenBank from France, USA and China. The second lineage (representing Slovenian OC43/SLO/60954/2015, KX059632) has a high nucleotide identity (99.74%) to OC43/Seattle/USA/SC0776/2019 (MN310478) and to available OC43 strains from France, USA and China. Two strains (OC43/SLO/61445/2016, KX059652 and OC43/SLO/62519/2016, KX059653) form the third lineage separate from the other two lineages and are most closely related (99.74%) to OC43/Seattle/USA/SC0776/2019 (MN310478) and several other strains from USA France and China (Figure 1).

From the determined 24 BCoV sequences, 12 originated from live cattle with respiratory symptoms, 7 from dead animals with pneumonia and 5 from dead animals with diarrhea (Appendix A). The sequenced 24 positive BCoV samples were collected from 18 different cattle herds, located in 15 different municipalities throughout Slovenia. BCoV were closely related to each other, with 99.2–100% nucleotide identity. The most closely related BCoV sequences from GenBank were FRA/EPI/Caen/2014 (KT318109) from France (100%) and AKS-01 (KU886219) from China (99.7%) and other CoV strains from different species (Figure 1). Human enteric coronavirus strain 4408 (FJ938067) and the identified Slovenian BCoV from this study share 99.7% nucleotide identity.

Determined BCoVs from cattle and HCoV-OC43 share 96.4–97.1% nucleotide and 96.9–98.5% amino acid identity. The most closely related human and bovine sequences from this study are HCoV-OC43/SLO/63863/2016 and BCoV/SLO/5580/2013, which share 97.1% nucleotide identity. The HCoV-OC43 and HCoV-HKU1 were the most prevalent in the whole study period (from February 2010 to February 2016), with increasing numbers of detected HCoV-HKU1 strains in 2014, 2015, and 2016, while the highest number of HCoV-OC43 strains were identified in 2014 (Figure 2).

## 4. Discussion

This study is the first genetic comparison of CoVs circulating in Slovenian human and cattle populations and their phylogenetic relationship with CoVs available in GenBank database. A phylogenetic comparison of 390 nucleotides long sequences of RdRp gene showed that BCoV and HCoV were clearly separated from each other, and further CoVs differentiation is evident; one species represents BCoV only, and three other previously known species represent HCoV-OC43, HCoV-HKU1, and HCoV-NL63. The identified BCoV (*n* = 24) and HCoV-OC43 (*n* = 31) share 96.4–97.1% nucleotide and 96.9–98.5% amino acid identity with each other, confirming a historically close relationship between BCoVs and HCoV-OC43, as described by Vijgen et al. [28]. Although Kin et al. demonstrated three sub-clusters within the BCoV cluster by using phylogenetic analysis of three genes (nsp12, S and N) with longer sequences [44] than in our study, the comparison of 390 nucleotides long sequences of RdRp not allows further sub-clustering of BCoV, as expected. Twenty-four BCoV sequences collected between 2012 and 2015 from 18 different cattle herds in this study were closely related to each other with 99.2% to 100% nucleotide identity confirming the identification of a genetically homogeneous group of BCoV in Slovenia, detected from live cattle with respiratory symptoms, dead animals with pneumonia, and dead animals with diarrhea. Nevertheless, even that rather small number of BCoV positive samples were sequenced, these BCoV positive samples were originated from fifteen different municipalities, which are from five of a total twelve different administrative regions in Slovenia. The results of this study thus represent important data about circulating field strains in cattle herds and the first genetic characterization of BCoV from our country. Within the Slovenian BCoV sequences, the closest published BCoV is FRA/EPI/Caen/2014 (KT318109) from France (100% nucleotide identity) and AKS-01 (KU886219) from China (99.7% nucleotide identity), confirming the identification of genetically identical or very similar BCoV strains than identified in some other countries worldwide. If more BCoV sequences on the RdRp gene would have been available also from other countries, the analysis could have been more precise also for this viral genome region.

In the whole study period, between February 2010 and February 2016, the detected CoVs from human and cattle patients were collected, with the highest observed prevalence in winter months, confirming strong seasonality for the detected positive samples as previously observed [45]. The genetic comparison of 66 HCoV and 24 BCoV from patients with clinical signs of respiratory disease did not provide evidence for zoonotic transmission of BCoV from bovine patients to humans in Slovenia. However, according to the close relationship between the HCoV and BCoV strains, we cannot completely exclude the possibility of interspecies transmission if only mild clinical symptoms are present in human patients because, in this case, patients will not be sampled and tested as in this study, in which only samples from patients with clear and severe respiratory signs or diarrhea were included. To better understand the epidemiology of closely related strains of CoV, longer sequences with complete genomes or specific genes (nsp12, S and N) will be needed from these archive samples and compared in a longer study period [14,15,23,44].

The previous study in Slovenia, with the testing of 592 samples of hospitalized children under six years, identified 40 HCoV positive samples between 2007 and 2008, and four HCoV species were detected using the real-time PCR method, with the highest prevalence of HCoV-HKU1 (52.5%), HCoV-OC43 (17.5%), 229E (15%) and NL63 (15%) [46]. A similar distribution of HCoV was identified in this study by direct Sanger sequencing of a 390-nt-long partial RdRp gene: HCoV-HKU1 (51.51%), HCoV-OC43 (46.96%), 229E (1.51%), suggesting the circulation of genetically similar strains of HCoV in the human population for several years. The obtained total of 90 new sequences from Slovenia in this study is the first genetic comparison of data for circulating field HCoV and BCoV, deposited in GenBank (KX059608–KX059697). As other researchers had assumed, we confirmed that the sequencing and phylogenetic analysis based on 390-nucleotide-long sequences of a single gene might not be sufficient to define the differences between genetically closely related strains, but this study confirmed that clear differentiation between different human, BCoV and other CoV is possible. This approach can be an important tool for the fast and reliable identification and characterization of genetically diverse CoVs, because the RdRp gene is a highly conserved region of the viral genome [30,47]. Although Lau et al. suggested that the more accurate phylogenetic analysis should be performed by amplification and sequencing of at least two gene loci, one from ORF1ab (e.g., RdRp or helicase) and one from HE to N (e.g., S or N) [30] this is not always possible for smaller laboratories. Sequencing of additional gene region was not done in our study, but genetic characterization of bovine and human CoV strains for differentiation of closely related strains from selected archive samples is still possible in further studies.

The whole seasons with sequenced CoV strains in our study were represented only in 2014 and 2015. In other years positive samples of CoV were included into sequencing because they were available in laboratory archive from routine testing. The observed high two-year dynamic for HCoV-OC43, which was the most prevalent HCoV in 2014 (14/19, 73.7%) and HCoV-HKU1, which was most frequently detected HCoV in 2015 (11/16, 68.7%) showed needs for continuously sequencing of field strains of CoV, also better understand the epidemiology of not highly virulent HCoVs. The majority of BCoV positive cattle samples were identified within the determination of the prevalence of ten pathogens (one of these pathogens was BCoV) detected by a real-time PCR method in live [27] and dead cattle with respiratory disease. Other BCoV positive samples were collected from individual detected positive cases, but rather low number of field samples are tested each year in cattle for this virus in our routine laboratories. Although positive samples of HCoV and BCoV were detected in all years of our study period, only limited of them were included further processed for sequencing. However, observations from this study provides also some important first data about circulation of CoV strains in human and cattle population in Slovenia and further studies are needed, based on more complete genome sequences and from a longer period of study.

The similarity between species within the *Betacoronaviruses* genus is high and, therefore, the interspecies transmission is a common phenomenon leading to the emergence of new pathogens, such as SARS-CoV and MERS-CoV [11,12]. Genetically closely related strains within the *Betacoronavirus 1* species, including BCoV, PHEV, Canine respiratory coronavirus (CRCoV) and HCoV-OC43 and close contacts between animals and humans, also provides possibilities for adaptation to the human host which can lead to a new human coronavirus, as happened at the end of the 19th century with HCoV-OC43 [44]. The most closely related human and bovine sequences from this study are HCoV-OC43 and BCoV, which share 97.1% nucleotide identity, both detected between 2013 and 2016. Due to CoV’s characteristics and close human-to-animal contact, the continuous epidemiological and phylogenetic surveys are needed to understand the epidemiology of these two genetically closely related CoVs.

In conclusion, the genetic comparison of 90 field positive samples of CoVs, circulating in Slovenian human and cattle populations, showed that 34 of them were grouped as HCoV-HKU1, 31 as HCoV-OC43 and one as HCoV 229E, while all 24 cattle positive samples were grouping only as BCoVs. The genetic comparison of determined strains of BCoVs and HCoV-OC43 revealed 96.4–97.1% nucleotide identity to each other, with clearly genetic differentiation between human and cattle CoV strains. Sequencing and phylogenetic analysis, based on 390-nucleotide-long sequences of RdRp gene, provide fast and reliable differentiation between different strains of HCoV, BCoV and also other CoV, but for better differentiation of genetically very closely related strains sequencing of additional virus genome region or sequencing of complete genome is needed. The result of this study provides the first genetic characterization of field HCoV and BCoV from Slovenia.

## Figures and Tables

**Figure 1 viruses-13-00676-f001:**
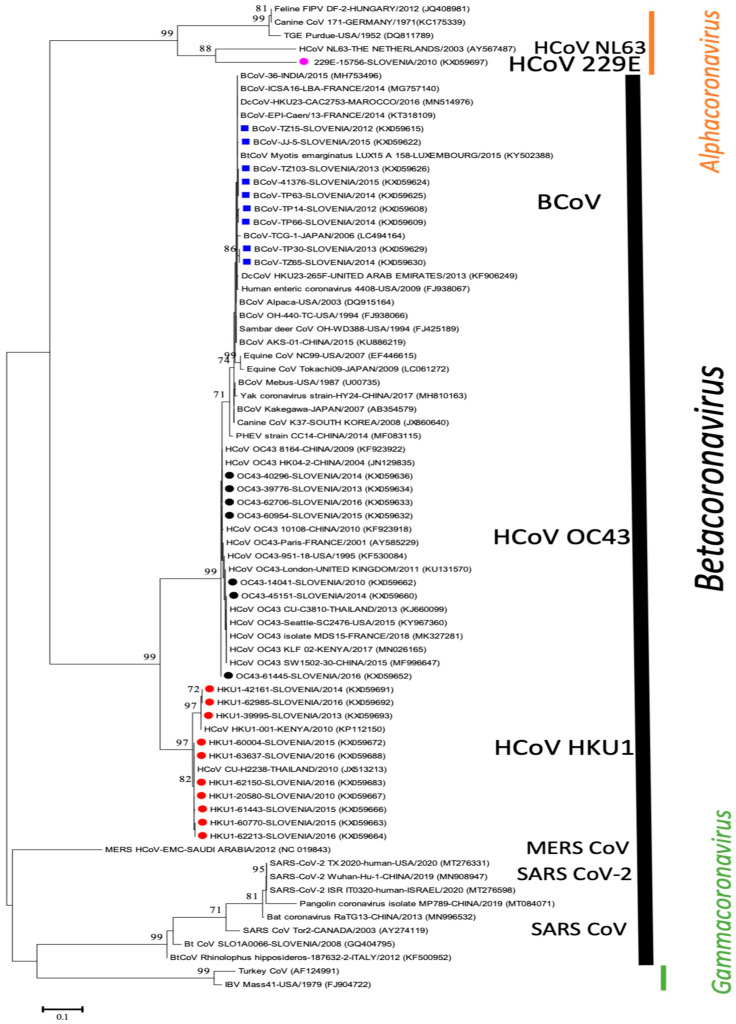
Phylogenetic tree based on 390-nucleotide-long sequences of RdRp gene with 9 representatives of Slovenian BCoV (■) and 18 representatives of Slovenian HCoV: HCoV-OC43 (●), HCoV-HKU1 (●), HCoV-229E (●), including 46 CoVs from GenBank database (with name of CoV strains, country, and accession numbers). Bootstrap values below 70 are not shown. Phylogenetic tree shows only representative sequences from Slovenia while the identified 100% identical sequences were not presented on tree because of graphical limits.

**Figure 2 viruses-13-00676-f002:**
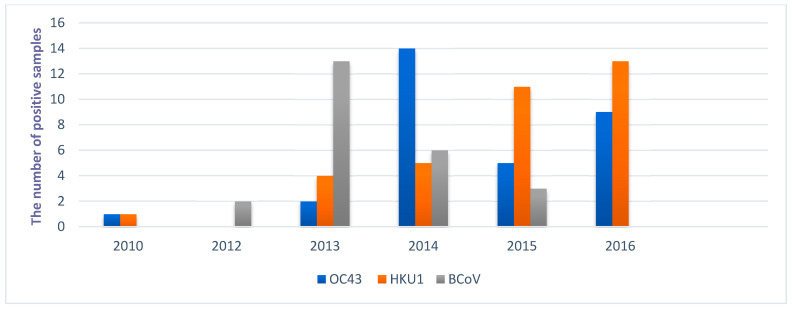
Yearly distribution of sequenced human coronavirus OC43 and HKU1 genotypes (HCoV-OC43 and HCoV-HKU1) and *Bovine coronavirus* (BCoV) in Slovenia during 2010 and 2016.

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
