# Peer review of "Genetic Characterisation and Comparison of Three Human Coronaviruses (HKU1, OC43, 229E) from Patients and Bovine Coronavirus (BCoV) from Cattle with Respiratory Disease in Slovenia"

_viruses, 2021, doi:10.3390/v13040676_

Round 1

Reviewer 1 Report

The manuscript “Genetic characterisation and comparison of three human coronaviruses (HKU1, OC43, 229E) from patients and bovine coronavirus (BCoV) from cattle with respiratory disease in Slovenia” presents the results of a 6-year long collection of human and bovine samples processed for the detection of respiratory coronaviruses. The manuscript is clear, well written, rich in bibliographic background and extensively discussed.

In my opinion, the manuscript thus needs only a few adjustments.

Minor revisions:

Materials and methods: line 151, the number of positive samples that have been sequenced should be considered as a result and moved to the specific section, while the 27-Ct criterium should be justified, for example due to desired/expected sequence quality or quantification of viral copies in the samples.

Results: it would be appropriate to show the sample distribution during the study. The authors should add the number of farms from which bovine samples were collected (that could also be discussed as a possible cause of strain similarity or support the thesis of a highly homogeneous group) and a description of HCoV-real time RT-PCR results for each year of the study.

Discussion: the discussion lacks of a proper conclusion paragraph and recombination is mentioned even if recombination events were not investigated in the current study. The authors should discuss eventually why this analysis was not performed and rearrange the paragraph.

Figure 1: the authors should specify how Slovenian HCoV and BCoV sequences were chosen as representative in the phylogenetic tree (were identical sequences collapsed for graphical reasons?)  

English syntax:

Line 50: “the novel SARS-CoV-2 was discovered” or “a novel coronavirus, SARS-CoV-2, was discovered”

Line 53: “from January 2020 to April 2021, SARS-CoV-2 was…”

Lines 248-250: “if more sequences would have been available…the analysis could have been more precise”

Line 249: “from other countries”

Line 260: “signs or…”  

Author Response

RESPONSE TO REVIEWERS

viruses-1176565

Reviewer 1

Ad1. Reviewer 1

The manuscript “Genetic characterisation and comparison of three human coronaviruses (HKU1, OC43, 229E) from patients and bovine coronavirus (BCoV) from cattle with respiratory disease in Slovenia” presents the results of a 6-year long collection of human and bovine samples processed for the detection of respiratory coronaviruses. The manuscript is clear, well written, rich in bibliographic background and extensively discussed.

In my opinion, the manuscript thus needs only a few adjustments.

Ad1. Authors’ response

We would like to thank the Reviewer for his/her general positive comment about our manuscript.

Ad2. Reviewer 1

Materials and methods: line 151, the number of positive samples that have been sequenced should be considered as a result and moved to the specific section, while the 27-Ct criterium should be justified, for example, due to desired/expected sequence quality or quantification of viral copies in the samples.

Ad2. Authors’ response

Thank you for your observation. In accordance with the Reviewer’s suggestion, the following sentence has been deleted from M&M:

“Ninety positive samples (66 of human and 24 of bovine origin) with cycle thresholds (Ct) below 27 were included in further direct Sanger sequencing process and phylogenetic comparison.”

Regarding the part of the sentence and 27-Ct criterium this part was deleted because in our opinion this is not important information in the manuscript, but already well known for laboratories that expected sequence quality is low/or sequencing is impossible when low copy number are present in PCR product.

The last sentence in this paragraph was changed into:

Line 158: The amplified products were detected via agarose gel electrophoresis and sequenced by Sanger sequencing, as described in a previous publication [43].

In the Results chapter, the following sentence was changed:

Line 183: To determine the overall genetic diversity, 66 HCoVs and 24 BCoV have been included in sequencing and phylogenetic analysis.

Ad3. Reviewer 1

Results: it would be appropriate to show the sample distribution during the study. The authors should add the number of farms from which bovine samples were collected (that could also be discussed as a possible cause of strain similarity or support the thesis of a highly homogeneous group) and a description of HCoV-real time RT-PCR results for each year of the study.

Ad3. Authors’ response

We agree with the Reviewer's suggestion.

The sample distribution data of this study were shown in supplementary data (S1), where all 90 determined CoV sequences in this study were presented with the date of samples collection, host, clinical observation name of sample and GenBank accession numbers.

In M&M the number of cattle herds, where 133 samples of affected live cattle and 84 dead cattle with pneumonia and diarrhea were collected.

In addition, two sentences were modified as follow:

Line 130: From 133 affected live cattle (collected from 24 different cattle herds) with symptoms of respiratory disease, nasopharyngeal swab samples were collected into sterile swabs (Sigma Virocult®, MW 951S, United Kingdom). From 84 dead cattle with pneumonia and/or diarrhoea originated from 76 different cattle herds, 10 cm³ lung tissue and/or faeces samples were collected.

Line 211: In the Results chapter the number of cattle herds, from which BCoV were successfully sequenced was added.

The sequenced 24 positive BCoV samples were collected from 18 different cattle herds, located in 15 different municipalities throughout Slovenia.

Ad4. Reviewer 1

Discussion: the discussion lacks a proper conclusion paragraph and recombination is mentioned even if recombination events were not investigated in the current study. The authors should discuss eventually why this analysis was not performed and rearrange the paragraph.

Ad4. Authors’ response

Thank you for your comment. We agree with the Reviewer and an additional paragraph with conclusions was prepared as follows:

Line 364:

In conclusion, the genetic comparison of 90 field positive samples of CoVs, circulating in Slovenian human and cattle populations, showed that 34 of them were grouped as HCoV-HKU1, 31 as HCoV-OC43 and one as HCoV 229E, while all 24 cattle positive samples were grouping only as BCoVs. The genetic comparison of determined strains of BCoVs and HCoV-OC43 revealed 96.4-97.1% nucleotide identity to each other, with clearly genetic differentiation between human and cattle strains. Sequencing and phylogenetic analysis, based on 390-nucleotide-long sequences of RdRp gene, provide fast and reliable differentiation between different strains of HCoV, BCoV and other CoV, but for better differentiation of genetically very closely related strains sequencing of additional virus genome region or sequencing of complete genome is needed. The result of this study provides the first genetic characterisation of field HCoV and BCoV from Slovenia.

Thank you for your comment regarding the paragraph about recombination, even if recombination events were not investigated.

These sentences were deleted from the last part of the discussion:

It seems that recombination between animal viruses and adaptation to the human host can lead to a new human coronavirus, as happened at the end of the 19th century with HCoV-OC43. Recombination within HCoV-OC43 is more frequent than within BCoV [44, 49-51]. In our research, we found only one cluster of BCoV and three clusters of HCoV-OC43.

Because these four references were deleted:

Morfopoulou, S.; Brown, J.R.; Davies, E.G.; Anderson, G.; Virasami, A.; Qasim, W.; Chong, W.K.; Hubank, M.; Plagnol, V.; Desforges, M.; Jacques, T.S.; Talbot, P.J.; Breuer, J. Human Coronavirus OC43 Associated with Fatal Encephalitis. The New England journal of medicine 2016, 375:497-498.

Lim, Y.X; Ng, Y.L.; Tam, J.P.; Liu, D.X. Human Coronaviruses: A Review of Virus-Host Interactions. Diseases 2016, 4.

Decaro, N.; Buonavoglia, C. An update on canine coronaviruses: viral evolution and pathobiology. Veterinary microbiology 2008, 132:221-234.

Erles, K.; Shiu, K.B.; Brownlie, J. Isolation and sequence analysis of canine respiratory coronavirus. Virus research 2007, 124:78-87.

Ad5. Reviewer 1

Figure 1: the authors should specify how Slovenian HCoV and BCoV sequences were chosen as representative in the phylogenetic tree (were identical sequences collapsed for graphical reasons?)

Ad5. Authors’ response

Thank you for your comment.

If all 90 CoV sequences from this study will be shown, the graphical presentation would not be possible.

The additional sentence was added to explain Figure 1:

Line 266:

The phylogenetic tree (Figure 1) shows only representative sequences from Slovenia while other 100% identical sequences were not presented on the tree because of graphical limits.

Ad6. Reviewer 1

Ad6. Authors’ response

Thank your English syntax comment. All your suggestions were accepted in the manuscript as follows:

Line 50: “the novel SARS-CoV-2 was discovered” or “a novel coronavirus, SARS-CoV-2, was discovered”

Response: The following sentence was changed from a novel SARS-CoV-2 to the novel SARS-CoV-2….

Line 53: “from January 2020 to April 2021, SARS-CoV-2 was…”

Response: done as suggested.  The article the has been deleted from the sentence.

Lines 288-298: “if more sequences would have been available…the analysis could have been more precise

Response: Two sentences were changed as suggested. 

Nevertheless, even that a rather small number of BCoV positive samples were sequenced, these BCoV positive samples were originated from 15 different municipalities, which are from 5 of total 12 different administrative regions in Slovenia. The results of this study thus represent important data about circulating field strains in cattle herds and the first genetic characterisation of BCoV from our country. Within the Slovenian BCoV sequences, the closest published BCoV is FRA/EPI/Caen/2014 (KT318109) from France (100% nucleotide identity) and AKS-01 (KU886219) from China (99.7% nucleotide identity), confirming the identification of genetically identical or very similar BCoV strains than identified in some other countries worldwide. If more BCoV sequences on the RdRp gene would have been available also from other countries, the analysis could have been more precise also for this viral genome region.

Line 297: “from other countries

Response: done as suggested. 

Line 308: “signs or…”

Response: done as suggested. 

We would like to thank the Reviewer for all his/her valuable comments to improve our manuscript.

Reviewer 2 Report

This manuscript describes a phylogenetical relationship between HCoV and BCoV collected in Slovenia from 2010 to 2016. This manuscript might be not acceptable in this journal, because of little novel information. If the authors attempt to analyze and present the remaining genes except RdRp using these samples, the revised manuscript including the additional data will be suitable for publication in this journal.

Major comments

Figure 1:

It is not clear. Please remake a new one.

You should show representative strains in a same style as follows: strain, collection country, collection year and GenBank accession numbers.

You should add the genera of strains used in this analysis on right of Figure.

Results

Page 4, line 187-Page 5, line 206 (4th -5th paragraphs):

Please explain a difference between genotypes and lineages you mentioned.

You should describe clear criteria to distinguish into genotypes (lineages).

Minor comments

Page 3, line 143:

You should not use abbreviated form of bacteria.

You should add ethics statement of samples collected from cattle.

Author Response

RESPONSE TO REVIEWERS

viruses-1176565

Reviewer 2

Ad1. Reviewer 2

This manuscript describes a phylogenetical relationship between HCoV and BCoV collected in Slovenia from 2010 to 2016. This manuscript might be not acceptable in this journal, because of little novel information. If the authors attempt to analyze and present the remaining genes except RdRp using these samples, the revised manuscript including the additional data will be suitable for publication in this journal.

Ad1. Authors’ response

We would like to thank the Reviewer for these valued comments about our manuscript.

The manuscript presents new data about 90 field positive CoV samples, which were genetically characterized for the first time in Slovenia. Genetically different strains of CoV were detected from human and cattle populations in Slovenia in the same geographic region and collection time. We think that these data provide important information not only for Europe but also for other parts of the world because a very limited number of accessible sequences in GenBank on this area and our geographic region is available. Our results showed that using this approach fast and reliable differentiation between HCoV-HKU1, HCoV-OC43, and HCoV-229E and BCoVs (and also other CoV) is possible and useful for diagnostic and research laboratories, dealing with CoV.

The information about the sequencing of other regions of viral genomes would not give us much better information about different strains of CoV, because RdRp gene, which part of them was sequenced in our study, is the most and highly conserved gene from Coronaviridae family. Of course, sequencing of another part of the genome or even a complete genome will be possible, but now we do not have the possibility for sequencing 90 positive CoV samples and no additional financial resources to continue with this, as you suggested. Shortcomings were discussed and corrected according to your suggestions in the manuscript and especially in the discussion chapter.

Ad2. Reviewer 2

Figure 1: It is not clear. Please remake a new one.

You should show representative strains in the same style as follows: strain, collection country, collection year and GenBank accession numbers. You should add the genera of strains used in this analysis on right of Figure.

Ad2. Authors’ response

We would like to thank the Reviewer for her/his comment. We accept the Reviewer suggestion.

Figure 1 was adopted as suggested by the Reviewer. New Figure 1 was prepared, with additional data for reference strains and genera of strains used in the analysis. 

Ad3. Reviewer 2

Results

Page 4, line 187-Page 5, line 206 (4th -5th paragraphs):

Please explain a difference between genotypes and lineages you mentioned.

You should describe clear criteria to distinguish into genotypes (lineages).

Ad3. Authors’ response

Thank you for your comment about the differentiation between genotypes and lineages.

The criteria for differentiation of CoVs, based on a different parts of the viral genome are still not fully clear and are used for different purposes in different manuscripts.

In our study, the 390 nt long sequences were compared with those in GenBank (using Blast and MEGA) and presented on phylogenetic tree. We did not have any problems identifying the right previously determined species for each CoV for all of the sequences determined in our study.

From the discussion part, we delete sentences with genotypes and we present only general differentiation of CoV according to previously known species of CoV (Figure 1).

We agree with the Reviewer. Several sentences were adopted, mainly in the discussion chapter as follow:

Line 280:

Although Kin et al. demonstrated three sub-clusters within the BCoV cluster by using phylogenetic analysis of three genes (nsp12, S, and N) with longer sequences [44] than in our study, the comparison of 390 nucleotides long sequences of RdRp not allows further sub-clustering of BCoV, as expected. Twenty-four BCoV sequences collected between 2012 and 2015 from 18 different cattle herds in this study were closely related to each other with 99.2% to 100% nucleotide identity confirming the identification of a genetically homogeneous group of BCoV in Slovenia, detected from live cattle with respiratory symptoms, dead animals with pneumonia, and dead animals with diarrhoea. Nevertheless, even that a rather small number of BCoV positive samples were sequenced, these BCoV positive samples were originated from 15 different municipalities, which are from 5 of total 12 different administrative regions in Slovenia. The results of this study thus represent important data about circulating field strains in cattle herds and the first genetic characterisation of BCoV from our country.

Line 328:

Although Lau et al. suggested that the more accurate phylogenetic analysis should be performed by amplification and sequencing of at least two gene loci, one from ORF1ab (e.g., RdRp or helicase) and one from HE to N (e.g., S or N) [30] this is not always possible for smaller laboratories. Sequencing of additional gene regions was not done in our study, but the genetic characterisation of bovine and human CoV strains for differentiation of closely related strains from selected archive samples is still possible in further studies. 

Line 335:

The whole seasons with sequenced CoV strains in our study were represented only in 2014 and 2015. In other years positive samples of CoV were included into sequencing because they were available in laboratory archive from routine testing. The observed high two-year dynamic for HCoV-OC43, which was the most prevalent HCoV in 2014 (14/19, 73.7%) and HCoV-HKU1, which was most frequently detected HCoV in 2015 (11/16, 68.7%) showed needs for continuously sequencing of field strains of CoV, also better understand the epidemiology of not highly virulent HCoVs. The majority of BCoV positive cattle samples were identified within the determination of the prevalence of ten pathogens (one of these pathogens was BCoV) detected by a real-time PCR method in live [27] and dead cattle with respiratory disease. Other BCoV positive samples were collected from individual detected positive cases, but a rather low number of field samples are tested each year in cattle for this virus in our routine laboratories. Although positive samples of HCoV and BCoV were detected in all years of our study period, only a limited of them were included further processed for sequencing. However, observations from this study provides also some important first data about the circulation of CoV strains in the human and cattle population in Slovenia, and further studies are needed, based on more complete genome sequences and from a longer period of study.

Ad4. Reviewer 2

Minor comments

Page 3, line 143:

You should not use abbreviated form of bacteria.

Ad4. Authors’ response

We would like to thank the Reviewer for her/his comment. The suggestion was accepted and corrected in the manuscript.

Line 164:

A commercial TaqMan® real-time PCR kit for the detection of seven major ruminant pathogens (LSI VetMAX™ Screening Pack – Ruminants Respiratory Pathogens, LSI, Lissieu, France), which allows the simultaneous detection of the Micoplasma bovis, Histophilus somni, Pasteurella multocida, Mannheimia haemolytica, BCoV, Bovine respiratory syncytial virus (BRSV), and Bovine parainfluenza 3 (PI-3) was used as previously described [27].

Ad5. Reviewer 2

You should add ethics statement of samples collected from cattle.

Ad5. Authors’ response

Thank the Reviewer for her/his comment. The suggestion was accepted and corrected in manuscript.

The following sentence has been included:

Line 109:

Ethics approval for testing animal specimens was not needed since samples were primarily taken for routine diagnostic surveillance by the local veterinary specialists.

We would like to thank the Reviewer for all his/her valuable comments, which help us to improve our manuscript.

Round 2

Reviewer 2 Report

This  manuscript has been improved according to reviewer's comments.

Thus, the revised manuscript is acceptable for publication in this journal.